# Exploring the Anticancer Potential of Proton Pump Inhibitors by Targeting GRP78 and V-ATPase: Molecular Docking, Molecular Dynamics, PCA, and MM-GBSA Calculations

**DOI:** 10.3390/ijms26178170

**Published:** 2025-08-22

**Authors:** Abdo A. Elfiky, Kirolos R. Mansour, Yousef Mohamed, Yomna Kh. Abdelaziz, Ian A. Nicholls

**Affiliations:** 1Biophysics Department, Faculty of Science, Cairo University, Giza 12613, Egypt; 2Academy of Scientific Research and Technology (ASRT), 101 Kasr Al-Ainy St., Cairo 11516, Egypt; 3Bioorganic & Biophysical Chemistry Laboratory, Linnaeus University Centre for Biomaterials Chemistry, Department of Chemistry & Biomedical Sciences, Linnaeus University, SE-39182 Kalmar, Sweden; ian.nicholls@lnu.se; 4Biotechnology/Biomolecular Chemistry Department, Faculty of Science, Cairo University, Giza 12613, Egypt; kirolosraafat465@gmail.com (K.R.M.); youssefmohmed992@gmail.com (Y.M.); yomnakhaled2002@gmail.com (Y.K.A.)

**Keywords:** GRP78, V-ATPase, drug repurposing, proton pump inhibitors, cancer cells, molecular docking, molecular dynamics simulation, principal component analysis, free-energy landscape, MM-GBSA

## Abstract

Cancer cells can adapt to their surrounding microenvironment by upregulating glucose-regulated protein 78 kDa (GRP78) and vacuolar-type ATPase (V-ATPase) proteins to increase their proliferation and resilience to anticancer therapy. Therefore, targeting these proteins can obstruct cancer progression. A comprehensive computational study was conducted to investigate the inhibitory potential of four proton pump inhibitors (PPIs), dexlasnoprazole (DEX), esomeprazole (ESO), pantoprazole (PAN), and rabeprazole (RAB), against GRP78 and V-ATPase. Molecular docking revealed high-affinity scores for PPIs against both proteins. Moreover, molecular dynamics showed favorable root mean square deviation values for GRP78 and V-ATPase complexes, whereas root mean square fluctuations were high at the substrate-binding subdomains of GRP78 complexes and the α-helices of V-ATPase. Meanwhile, the radius of gyration and the surface-accessible surface area of the complexes were not significantly affected by ligand binding. Trajectory projections of the first two principal components showed similar motions of GRP78 structures and the fluctuating nature of V-ATPase structures, while the free-energy landscape revealed the thermodynamically favored GRP78-RAB and V-ATPase-DEX conformations. Furthermore, the binding free energy was −16.59 and −18.97 kcal/mol for GRP78-RAB and V-ATPase-DEX, respectively, indicating their stability. According to our findings, RAB and DEX are promising candidates for GRP78 and V-ATPase inhibition experiments, respectively.

## 1. Introduction

Glucose-regulated protein 78 kDa (GRP78), also known as binding immunoglobulin protein (BiP), is synthesized by the *HSPA5* gene and belongs to the HSP70 family of heat shock proteins [1]. It serves a crucial role in all aspects of maintaining the balance of cellular proteins, such as folding, assembly, degradation, and transport across membranes [2]. The GRP78 structure consists of two domains: a nucleotide-binding domain (NBD), where adenosine triphosphate (ATP) or adenosine diphosphate (ADP) binds at the N-terminus, and the substrate binding domain (SBD) at the C-terminus [3]. Generally, GRP78 is found in the lumen of the endoplasmic reticulum (ER); it can also be partially transported to the cytoplasm, nucleus, mitochondria, cell membrane surface, or even released outside the cell in response to stress or damage [1]. GRP78 is an important regulator of the unfolded protein response (UPR) as it attaches to the luminal binding site of three receptors located in the ER, namely inositol-requiring kinase 1 (IRE1), activating transcription factor 6 (ATF6), and PKR-like eukaryotic initiation factor 2α kinase (PERK), to inhibit their activation unless unfolded or misfolded proteins are accumulating in the ER, triggering GRP78 translocation [4,5].

Vacuolar-type ATPase (V-ATPase) is a proton pump protein that is localized in many intracellular compartments, such as lysosomes, endosomes, the Golgi apparatus, and vesicles, for organelle acidification or in the cell membrane of some specialized cells that play a role in bone resorption, sperm maturation, and acid excretion for extracellular acidification [6,7]. V-ATPase consists of the cytoplasmic catalytic region V_1_, in which ATP hydrolysis takes place, and the membrane-embedded region V_0_, driving the transfer of protons for acidification [6,7,8]. V-ATPase is associated with many diseases, including encephalopathy [9], cutis laxa [10], and osteopetrosis [11], and was found to be involved in many cancer cases as well [12,13]. Interestingly, V-ATPase was found to be involved in many cellular pathways, such as mTOR [14], Notch [15], and Wnt/β-Catenin [16] pathways. The involvement of V-ATPase in these pathways and with cancer progression, metastasis [17], and increasing drug resistance has raised interest in V-ATPase inhibition as a potential treatment for many cancer variants [7,12,13].

Most cancer cells have elevated glycolytic activity with severe hypoxia, an acidic microenvironment, and glucose deprivation, leading to misfolded protein buildup in the ER, activation of the UPR, and consequently, overexpression of GRP78 on the cancer cell surface [18,19]. Cell surface GRP78 is a versatile receptor that binds to a range of different substances to create complexes that mediate different signaling pathways including phosphoinositide 3-kinase, protein kinase B [18,20], mammalian target of rapamycin (mTOR), mitogen-activated protein kinase, and NF-κB pathways; thus, it can influence many oncogenic activities, such as angiogenesis, invasion, migration, proliferation, and responses to chemotherapy and radiotherapy [21]. Moreover, cytosolic GRP78 is translocated to the lysosomal lumen in cancer cells to protect them against lysosomal proteases by stabilizing the lysosomal membranes [22], where V-ATPase is extensively expressed to preserve the pH gradients of the cancer cells’ lysosomes [13]. However, exposing cancer cells to further ER stress can selectively trigger apoptosis in these cells [23].

Proton pump inhibitors (PPIs) are a class of drugs that are used effectively for targeting gastric and peptic-acid-related diseases, such as peptic ulcer, gastric ulcer, gastroesophageal reflux disease, Zollinger–Ellison syndrome, and *Helicobacter pylori* infection [24,25,26]. PPIs target these diseases by inhibiting the synthesis of gastric and peptic acids [24,25]. PPIs inhibit acid synthesis by inhibiting the function of H^+^/K^+^ ATPase proteins by competing with ATP on the active site [26,27]. A study by Wei-ping demonstrated lysosomal stress induced by PPIs [28]. Furthermore, the study revealed the effect of V-ATPase and H^+^/K^+^ ATPase inhibition, which causes increasing lysosomal stress, ER stress, and reactive oxygen species inside cells [28]. PPIs activated the translocation of transcription factor EB during lysosomal stress after inhibition of V-ATPase and ER stress, which was identified by measuring the degree of expression of three major ER stress proteins (i.e., PERK, IRE1, and ATF6) in GES-1 cells (human gastric epithelial cell line) after incubation with PPIs, which showed increased expression of these proteins, highlighting the ER stress caused by PPIs [28].

Motivated by their common ATP-binding functions, this study aims to explore the potential of PPIs in targeting GRP78 and the stability of the V-ATPase–PPI complexes. This has been undertaken using a series of in silico studies, including molecular docking followed by 100 ns molecular dynamics. This study provides a prospective view on the possibility for the repurposing of PPIs for the co-targeting of GRP78 and V-ATPase and, consequently, the associated cancer.

## 2. Results and Discussion

The survival of cancer cells through metabolic and microenvironmental signals has raised challenges in the development of anticancer therapy [29]. During ER stress, cancer cells regulate the UPR by overexpressing GRP78 [30], which aids in their resistance to stress conditions and anticancer therapy [31]. Cancer cells can release excess acid into their extracellular environment by increasing V-ATPase expression. Consequently, they promote cancer cell proliferation and multidrug resistance, hindering apoptosis [32]. Therefore, inhibiting both V-ATPase [32] and GRP78 is a promising approach for cancer therapy [31]. Investigating PPI interactions with V-ATPase and GRP78 offers a possible approach for inhibiting both proteins.

### 2.1. Molecular Docking and Interactions

The inhibition potential of PPIs, particularly dexlansoprazole (DEX), esomeprazole (ESO), pantoprazole (PAN), and rabeprazole (RAB), against GRP78 and V-ATPase was primarily evaluated by comparing their binding affinities for the active site of the ATPase domain of the two proteins to that of ATP, the positive control. According to the results presented in Figure 1, DEX has the highest binding affinity scores of −10.8 and −8.6 kcal/mol for GRP78 and V-ATPase, respectively, followed by ESO, PAN, and RAB, revealing its high inhibition potential. Notably, PPIs showed higher binding affinity scores for GRP78 than V-ATPase. Moreover, the affinity binding scores of the PPIs were close to those of ATP.

PPIs can engage in various types of molecular interactions with the active sites of both GRP78 and V-ATPase, including conventional hydrogen bonds and carbon–hydrogen bonds, as well as anion/cation–π and π–alkyl interactions (Figure 2 and Figure 3). Furthermore, DEX and PAN can form halogen bonds due to the presence of fluorine atoms in their chemical structure, as seen in the case of fluorine atom number 24 in DEX, which forms a halogen bond of 3.12 Å with ASP224 in GRP78. The amino acid residue PHE252A, which is located in the ATPase domain of V-ATPase, was involved in the π-σ interactions with ESO and DEX, as well as the π-π interactions with PAN and RAB. In contrast, the amino acid residue SER365 of GRP78 contributes to van der Waals interactions with ESO and RAB. PPIs were observed to have more hydrophobic interactions with the proteins than ATP, while the latter can form more hydrogen bonds; however, it has some unfavorable bonding interactions with V-ATPase.

### 2.2. Molecular Dynamics Simulation

#### 2.2.1. RMSD Analysis

The root mean square deviation (RMSD) measures the stability of the complexes by accounting for the average variation in atomic position relative to the reference structure. The gmx_rms tool was used to calculate RMSD values, as seen in Table 1. ATP, DEX, and PAN showed similar structural deviation to the reference apo GRP78, while ESO and RAB demonstrated higher RMSD values due to the conformational changes in SBDα and SBDβ, which are especially seen in the GRP78-RAB complex during the 24–36 nm range of the simulation, as seen in Figure 4a, and conformational changes in SBDα in the GRP78-ESO complex after the 60 ns hallmark. Furthermore, all GRP78 structures, except for GRP78-ESO, retained a stable conformation starting at 40 nm in the simulation. For V-ATPase and its complexes, ATP binding enhanced the stability of the complex, while DEX, ESO, and PAN showed similar results to the apo protein structure. Furthermore, RAB elevated complex instability, as illustrated in Figure 4c, although it was rapidly unbound from the binding site. With the earlier release of ESO and the misallocation of PAN from the binding pocket during the simulation, GRP78-DEX demonstrated the most stable structure for binding site complexation.

Following the RMSD fluctuation of the ligands and VMD trajectory visualization, most ligands maintained their initial structure when bound to the active site, with RMSD fluctuations lower than 0.2 nm, (Figure 4b,d). It was also noted that most ligands were released after different time intervals from the binding site in GRP78: ~16 ns for DEX, ~32 ns for ESO, and ~60 ns for PAN, as seen by their increased RMSD fluctuation ranges and values. In the case of V-ATPase, release was observed after ~10 ns for ESO, ~50 ns for PAN, and ~37 ns for RAB. RAB and DEX maintained their active site binding in GRP78 and V-ATPase, respectively, with a slight deviation of RAB from its initial conformation.

#### 2.2.2. RMSF Analysis

The GROMACS built-in tool, gmx rmsf, was utilized to calculate the root-mean-square fluctuation, which is a measure of the degree of mobility/rigidity of residues and provides insights concerning complex stability or conformational changes. The average RMSF values for GRP78 and its complexes (-ATP, -DEX, -ESO, -PAN, and -RAB) were ~0.27 nm, ~0.25 nm, ~0.31 nm, ~0.32 nm, ~0.25 nm, and ~0.32 nm, respectively. The PAN complex shows greater stability comparable to ATP, while DEX, ESO, and RAB increased the overall mobility of their respective complexes, notably with a clear difference in mobility of the SBDα and SBDβ domains, as seen in Figure 5A at regions of residues 483–490 and 561–608, respectively. Thus, the increased mobility of the SBD could potentially inhibit protein action through the instability of peptide binding. The average RMSF values for V-ATPase and its complexes (-ATP, -DEX, -ESO, -PAN, and -RAB) were ~0.18 nm, ~0.17 nm, ~0.22 nm, ~0.20 nm, ~0.19 nm, and ~0.24 nm, respectively, for chain A and ~0.19 nm, ~0.19 nm, ~0.21 nm, ~0.18 nm, ~0.21 nm, and ~0.20 nm, respectively, for chain D. The unbound and ATP-bound V-ATPases had a lower degree of mobility, which is indicative of the higher stability of the chains throughout the 100 ns simulation. The V-ATPase–DEX, –ESO, and –RAB complexes showed higher fluctuation behavior than the V-ATPase-ATP complex, where the increased RMSF was mostly due to the rapid mobility of the alpha helices at the C-terminal of both chains, which is seen at the residue range of 473–616 (Chain A Figure 5B) and 404–506 (Chain D Figure 5C) and indicative of the instability of key components for ATP hydrolysis. The average binding site key residue fluctuations are documented in Table 1. For GRP78, ATP and PAN exerted stronger restrictive action on those residues, while DEX, ESO, and RAB did not, which might be due to the faster release of DEX and ESO from the binding pocket and the slight conformation change observed for RAB. For V-ATPase, the averages showed a similar trend to the overall RMSF, with notably more restrictive action of DEX, while PAN and ESO were disregarded as they were rapidly released from the binding site. This was in agreement with the previous observations of the different degrees of PPIs in suppressing V-ATPase activity [28].

#### 2.2.3. RoG and SASA Analyses

The radius of gyration (RoG) of protein–ligand complexes is a parameter that can be derived from MD simulations that reflect the stability of the protein–ligand complex, protein folding, and compactness [33,34]. The average RoG values were analyzed over the 100 ns MD simulations of the targeted protein complexes, as seen in Table 1. The RoG fluctuation of the different protein complexes was minor and similar to that of the apo proteins [33]. In the case of GRP78, ligands have the same pattern during MD simulations (Figure 6a,c). However, in the V-ATPase case, no significant difference in pattern was observed for all ligands studied, as compared to the protein–ATP complex that exhibits the stability of protein–ligand complexes.

The solvent-accessible surface area (SASA) was measured during the MD simulations, as it identifies the area exposed for solvation and the free environment [35], with average values for ligands close to those of the V-ATPase–ATP complex, as shown in Table 1. In Figure 6b,d, SASA analysis revealed that there was a gradual and slight increase in SASA values in GRP78 throughout the 100 ns simulation, which reflects the stability of the GRP78–ligand complexes due to the low variation in the values of ligands in contrast to the GRP78–ATP complex. Moving to V-ATPase, it showed different patterns for DEX and ESO than for the V-ATPase–ATP complex, and the other ligands showed different patterns during different periods throughout the 100 ns simulation, but the average values of the ligands were close to those of the V-ATPase–ATP complex.

The RoG and SASA results align together for both the GRP78 and V-ATPase systems. In the case of GRP78 complexes, they exhibited approximately similar values to the GRP78–ATP complex, indicating the stability of the GRP78-PPIs interaction. The V-ATPase case exhibited non-significantly higher values for DEX and RAB relative to V-ATPase-ATP and other V-ATPase-PPI complexes during RoG analysis and non-significantly higher values for DEX and ESO during SASA analysis. This indicates that protein stability was preserved after the interaction with the different PPIs [36]. The increasing values of SASA and RoG during the V-ATPase case are due to the structure of the V-ATPase protein, as it consists of two chains: chain A and chain D.

#### 2.2.4. Hydrogen Bond Analysis

To assess the impact of hydrogen bonds on the protein–ligand complex stabilities, the built-in gmx hbond tool was used to quantify hydrogen bonding with a hydrogen-donor–acceptor cutoff angle of 30° and a donor–acceptor cutoff distance of 0.35 nm. The number of observed hydrogen bonds of each protein–ligand complex is depicted in Figure 7, while their average hydrogen bonding is shown in Table 1. PAN showed the highest average among the simulated PPIs, although it was released from the binding pocket of GRP78 at ~60 ns. RAB, which is slightly delocalized from the binding pocket of GRP78, had a low average of hydrogen bonding, suggesting a significant contribution of other types of interactions to the stability of the binding of the PPIs to the proteins. For V-ATPase, DEX had the highest value among the PPIs; however, the value was significantly lower when compared to the control ATP, a similar trend to that for the GRP78 complexes.

#### 2.2.5. Principal Component Analysis

The high dimensionality of MDS data poses a challenge in understanding correlations in the data and interpreting the significance of movements and changes. PCA can be used to reduce the number of dimensions and effectively extract meaningful interpretations from the data. Principal component analysis was performed on the first two eigenvectors as they accommodate most of the information from the MD trajectories, accounting for more than 70% of the variance, with relatively lower values for the remaining vectors, as seen in Figure 8. MD trajectories were projected on PC1 and PC2, which show that GRP78 complexes occupy the same subspace, as seen in Figure 9A, with the exception of the GRP78-DEX complex (red). The converged subspace highlights the similar motion occurring to the complexes and their similarity to the GRP78-ATP complex, especially for the GRP78-PAN (green) and -RAB (blue) complexes, as they show a more compact subspace similar to the GRP78-ATP complex (black). The different occupied subspace region of the GRP78-DEX complex highlights the different conformational changes undergoing in the complex, which might be due to the instability and divergence of DEX from the active site. For V-ATPase structures, the projection subspaces were generally confined to a smaller region, although the projections were not well superimposed over one another (Figure 9B). This highlights the fluctuations of the different ligands, which are especially seen in the GRP78-RAB (blue) and -ESO (yellow) complexes, and is due to their release from the active site. The GRP78-DEX (red) and -PAN (green) complexes offer a more compacted region, similar to the V-ATPase complex, although the slightly different projections are indicative of conformational changes relative to that of the ATP complex (black). Conclusively, the complexes of DEX-GRP78 (red) and RAB-VATPase (blue) have deviated values compared to the ATP complexes (black).

#### 2.2.6. Free-Energy Landscape

Gibb’s free-energy-landscape (FEL) analysis offers an illustration of the thermodynamics of the complex, the stability of conformations, and the boundaries of transitions between conformations. The PC1 and PC2 clusters were used to construct FEL plots (Figure 10 and Figure 11). A single basin was observed for GRP78 structures (Figure 10), although the fluctuating nature of its GRP78-DEX and -ESO complexes resulted in an increased maximum ∆G value compared to that of the GRP78-ATP complex and minima conformers, where the ligand occupies a binding site other than the active site. The GRP78-PAN complex shows the closest resemblance to the GRP78-ATP complex, while for GRP78-RAB, although having a higher maximum ∆G value compared to the GRP78-ATP complex due to the short-lived conformer, it has more localized minimal energy, supporting the stability of the GRP78-RAB complex. For V-ATPase FEL analysis, multiple basins were seen for the V-ATPase-ATP, -RAB, and -ESO complexes, and the transitions between these different basins were through low-energy-level barriers (Figure 10). In the case of the widest basin of the V-ATPase-RAB complex, RAB was confined in the active site of the V-ATPase, suggesting the moderate stability of the V-ATPase-RAB complex. In contrast, ESO was not bound to the protein during its minima. For the V-ATPase-PAN complex, a single highly compacted basin was observed, although it was not when PAN was bound to the active site, suggesting significant stability and binding to the new binding site. Furthermore, V-ATPase-DEX achieved a single minimal basin after approximately 55 ns of the simulation with a high energy barrier between the basin and the intermediate conformer, suggesting the high stability of the basin conformation.

#### 2.2.7. Binding Free Energy

Molecular Mechanics–Generalized Born Surface Area (MM-GBSA) is an established method for determining protein–ligand binding energies and for identifying the amino acids contributing to the binding interactions [33,37,38]. MM-GBSA analysis was performed after protein complex equilibration from 40 ns to 100 ns for GRP78 and V-ATPase with different ligands, as shown in Figure 12. MM-GBSA demonstrated the stability of ATP, DEX, and RAB with GRP78, but the ESO complexes’ free energy showed values around zero for intervals over the 100 ns run (Figure 12a). The increasing values of binding free energy of the GRP78-PAN and GRP78-RAB complexes were attributed to the diffusion of these ligands from their original binding site. Moving to V-ATPase MM-GBSA shows the stability of V-ATPase–ligand complexes due to its negative binding free energy values, but the V-ATPase-DEX complex showed increasing values of binding free energy during the 100 ns run (Figure 12b).

An assessment of the contributions from van der Waals interactions (Table 2) revealed them to be the primary factor underlying the stability of protein interactions with the PPI systems. The data aligns with the free binding energies that were calculated in the MM-GBSA studies. The low impact of electrostatic interactions was attributed to the antagonism between electrostatic energy (EEL) and electrostatic contribution to the solvation free energy calculated by GB (EGB), gas-phase energy (GGAS), and solvation free energy (GSOLV), especially for V-ATPase-ATP complexes, explaining the weak total binding energy of ATP to V-ATPase, as compared to other complexes.

#### 2.2.8. Amino Acid Contribution

For GRP78 complexes, MM-GBSA analysis revealed that the amino acid residues TYR39 and ILE61 contributed significantly to the total binding energy of these complexes throughout the simulations, except for the GRP78-DEX complex (Figure 13). The residues making major contributions to the GRP78-DEX interactions were located in the 300–400 region of the protein, including the positively charged ARG324 and LYS370. The LYS96 residue significantly contributed to the stability of the GRP78-ATP complex, along with threonine residue numbers 38, 37, and 229 and glycine residue numbers 36, 227, and 228. This was similar to the multiple glycine residues that contributed to the interactions in the GRP78-PAN and GRP78-RAB complexes, in particular glycine residue numbers 226, 227, 363, and 364.

In the V-ATPase complexes (Figure 14), the amino acid residues contributing to binding were found in the 250–259 and 400–450 regions of chain A, with the residues ALA251A and PHE252A in common, except in the case of the V-ATPase-ESO complex. The PHE252A residue is mainly responsible for the hydrophobic interactions of the complexes. Notably, many of the positively charged arginine residues took part in the molecular interactions of the V-ATPase complexes, namely the ARG400D residue in the ATP, DEX, and RAB complexes and ARG442A, as well as ARG494D in the V-ATPase-ESO complex.

Generally, there were more contributing amino acid residues in the V-ATPase complexes than in the GRP78 complexes; however, the total binding energies of the GRP78 complexes indicate that they were of higher stability.

## 3. Materials and Methods

### 3.1. Structural Retrieval of Proteins and Ligands

The three-dimensional (3D) structures of the proteins were downloaded from the Research Collaboratory in Structural Bioinformatics Protein Data Bank (RCSB PDB) (https://www.rcsb.org/, Accessed on 10 July 2024). For GRP78 (PDB ID: 6ASY, resolution: 1.85 Å), only chain A was kept from this structure, which incorporates a full wild-type NBD compared to all previously identified Hsp70-ATP structures [2]. The full-length protein of human V-ATPase (PDB ID: 6WM3, resolution: 3.40 Å) was obtained to isolate chains A and D of the V_1_ complex, where ATP hydrolysis occurs [6]. The 3D conformers of ATP (PubChem CID: 5957) and the PPIs dexlansoprazole (PubChem CID: 9578005), esomeprazole (PubChem CID: 9568614), pantoprazole (PubChem CID: 4679), and rabeprazole (PubChem CID: 5029) were retrieved from the NCBI PubChem database (https://pubchem.ncbi.nlm.nih.gov/, Accessed on 10 July 2024) in structure data file (SDF) format. To prepare the PPIs for the AutoDock Vina docking tool, these SDF format files were converted to PDB format using the PyMOL 2.5.4 software [39].

### 3.2. Molecular Docking

Molecular docking techniques are used to assess the interaction between proteins and ligands [40] and to study the inhibition potentials of PPIs against V-ATPase and GRP78 proteins. AutoDock Tools 1.5.6 was used for the preparation of proteins and ligands for the molecular docking process [41,42]. The ligands’ PDBQT files were prepared for flex docking by making all possible rotatable bonds in a rotatable state. On the other hand, the proteins’ PDBQT files were prepared using the same tools by making the active site residues, such as the ASP34, THR37, THR38, TYR39, LYS96, GLU201, ASP224, LYS225, THR229, ASP231, GLU256, GLU293, LYS296, ARG297, SER300, SER365, ARG367, ILE368, and ASP391 residues of chain A in GRP78 (6ASY) and the PHE252, CYS254, LYS256, THR257, GLU283, PHE445, GLN522, ASN523, and TYR525 residues of chain A, with the ARG400 and LYS403 residues of chain D in V-ATPase (6WM3) in the flex state. Moreover, Kollman charges and polar hydrogens were added, while water molecules were removed. Then, grid boxes were customized for GRP78 centered at (9.462, 2.281, −0.515) Å, and the box size was (30 × 24 × 24 Å^3^), while for V-ATPase, the box was centered at (138.821, 165.124, 194.997) Å, and the box size was adjusted to be (24 × 32 × 24 Å^3^) with 1 Å spacing points covering the active site. The AutoDock Vina 1.1.2 software was used for different orientations of the ligands with the proteins, depending on free energy (ΔG) [43]. After the docking process, the different orientations of different proteins with the ligands were visualized using the PyMOL software.

### 3.3. Molecular Dynamics (MD) Simulations

The solvated systems were prepared using the CHARMM GUI 3.8 solution builder tool for all complexes and the apo proteins [44,45]. The parameters and topologies of the drugs were automatically configured using the CHARMM general force field (CGenFF) [46,47], and all histidine residues were manually inputted as HSP. A water box of the TIP3P water model was created at sizes of (151.8 × 132.7 × 92.7 Å^3^) and (137.9 × 99.5 × 89.9 Å^3^) for 6WM3 and 6ASY, respectively, at a pH of 7.0, and the systems were neutralized with a concentration of 0.154 M NaCl through the Monte Carlo placement method. All MD simulations were run using GROMACS/2021.3-foss-2021a-CUDA-11.3.1-PLUMED-2.7.2 [48] while employing the CHARMM36m force field [49]. The systems were minimized for 5000 steps using the steepest descent algorithm, followed by an NVT ensemble equilibration run of 125 ps using the Nose–Hoover algorithm at 310.15 K. The production runs were then performed for 100 ns with the same conditions applied in the equilibration run.

### 3.4. Principal Component Analysis (PCA) and Free-Energy-Landscape Analysis

Principal component analysis is a multivariate analysis tool that can be used for molecular dynamics data reduction by using MD simulation trajectories and evaluating internal coordinates and mass-weighted cartesian of the simulations [50,51]. The primary motion of the apo proteins and protein–ligand complexes was identified by eigenvalues using the diagonalization of covariance matrix projections and principal components (PC1 and PC2) [51,52]. Covariance matrices, eigenvalues, and eigenvectors were calculated using the gmx covar and gmx anaeig command lines. PCA was performed for C-α atoms of the polypeptide after removing the unfolded, highly flexible residues from the C-terminal of 6asy (residues 1–7) and 6wm3 (chain A: residues 1–4, chain D: residues 1–6) to decrease the randomness and free movement of the protein. The thermodynamics of the system was visualized by the Gibbs free-energy landscape, which was acquired by utilizing the projection of the trajectory on the eigenvectors with the gmx sham built-in tool.

### 3.5. Binding Free Energy Calculations

Molecular mechanics–generalized Born surface area (MM-GBSA) is an approach to compute the theoretical binding free energy of the ligands to the protein [53]. The gmx_MMPBSA 1.6.2 package was utilized within GROMACS to calculate the ∆G_binding_ energy by applying the MM-GBSA method [54,55]. The calculations were applied to the MD trajectories with a frame per 100 ps over the range of 40 ns to 100 ns (an equilibrated region of the trajectories) using the ∆G_binding_ energy equation as follows:∆G_binding_ = G_complex_ − (G_protein_ + G_ligand_)(1)
where G_complex_ is the protein–ligand complex total free energy and G_protein_ and G_ligand_ are the total free energy of the solvated protein and ligand, respectively.

### 3.6. Visualization and Data Analysis

BIOVIA Discovery Studio visualizer v24.1 was used to identify protein–ligand molecular interactions of the docked structures. Following MD simulations of the proteins and their complexes, the resulting trajectories were visualized with the Visual Molecular Dynamics (VMD) 1.9.3 software [56]. Structural parameters, including the root mean square deviation (RMSD), the root mean square fluctuation (RMSF), the surface-accessible surface area (SASA), the radius of gyration (RoG), the number of hydrogen bonds, the principal component, and the free-energy landscape, were estimated based on these trajectories throughout the simulation using GROMACS analysis tools. These data were plotted afterward using XMGRACE 5.1.25.

## 4. Conclusions

In this study, the interaction of the PPIs, namely pantoprazole (PAN), rabeprazole (RAB), esomeprazole (ESO), and dexlasnoprazole (DEX), with the cancer-related proteins GRP78 and V-ATPase were thoroughly computationally evaluated. Our molecular docking and molecular dynamics studies provided a basis for identifying DEX and RAB as propitious drugs to be experimentally tested for V-ATPase and GRP78 inhibition, respectively. Both the DEX-V-ATPase and RAB-GRP78 complexes demonstrated stable binding free energy values of −16.59 ± 0.22 and −18.97 ± 0.28 kcal/mol, respectively, and favorable thermodynamics of the conformers. In addition, PCA and MM-GBSA analyses provided insights into the roles of interaction type and identified key amino acid residues in the two proteins. Furthermore, the results obtained for RAB and PAN suggest that they may serve as bases for co-targeting both these proteins. Finally, these findings highlight the potential for repurposing established PPIs to target V-ATPase and GRP78, which in turn motivates in vitro and in vivo studies on the anticarcinogenic effect of PPIs.

## Figures and Tables

**Figure 1 ijms-26-08170-f001:**
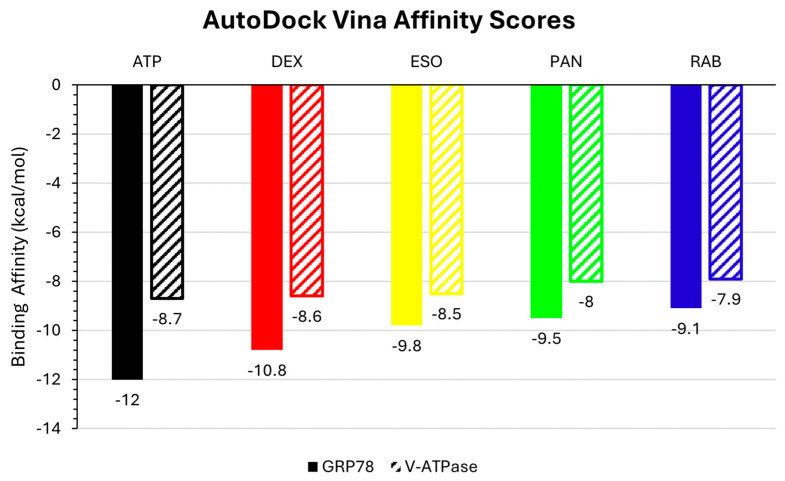
AutoDock Vina binding affinity scores of the ligands (DEX, ESO, PAN, RAB, and ATP) against GRP78 and V-ATPase.

**Figure 2 ijms-26-08170-f002:**
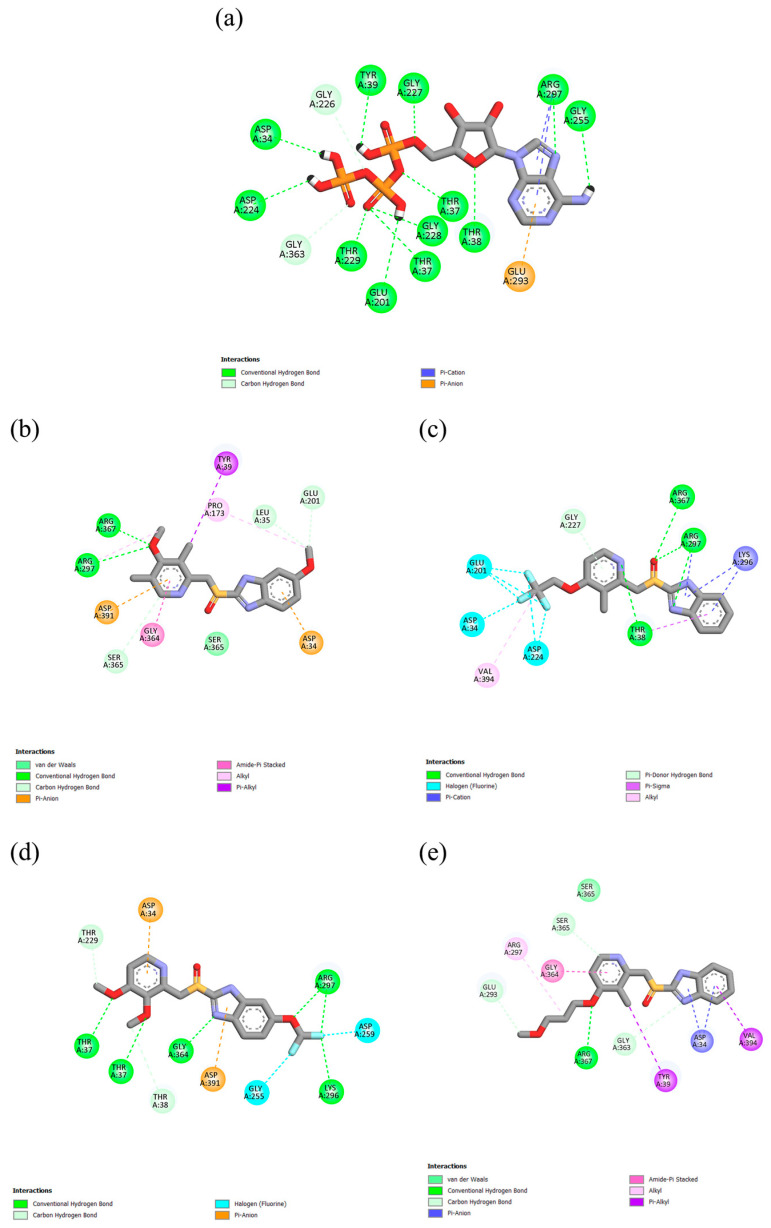
Molecular interactions of (**a**) ATP, (**b**) ESO, (**c**) DEX, (**d**) PAN, and (**e**) RAB with human GRP78.

**Figure 3 ijms-26-08170-f003:**
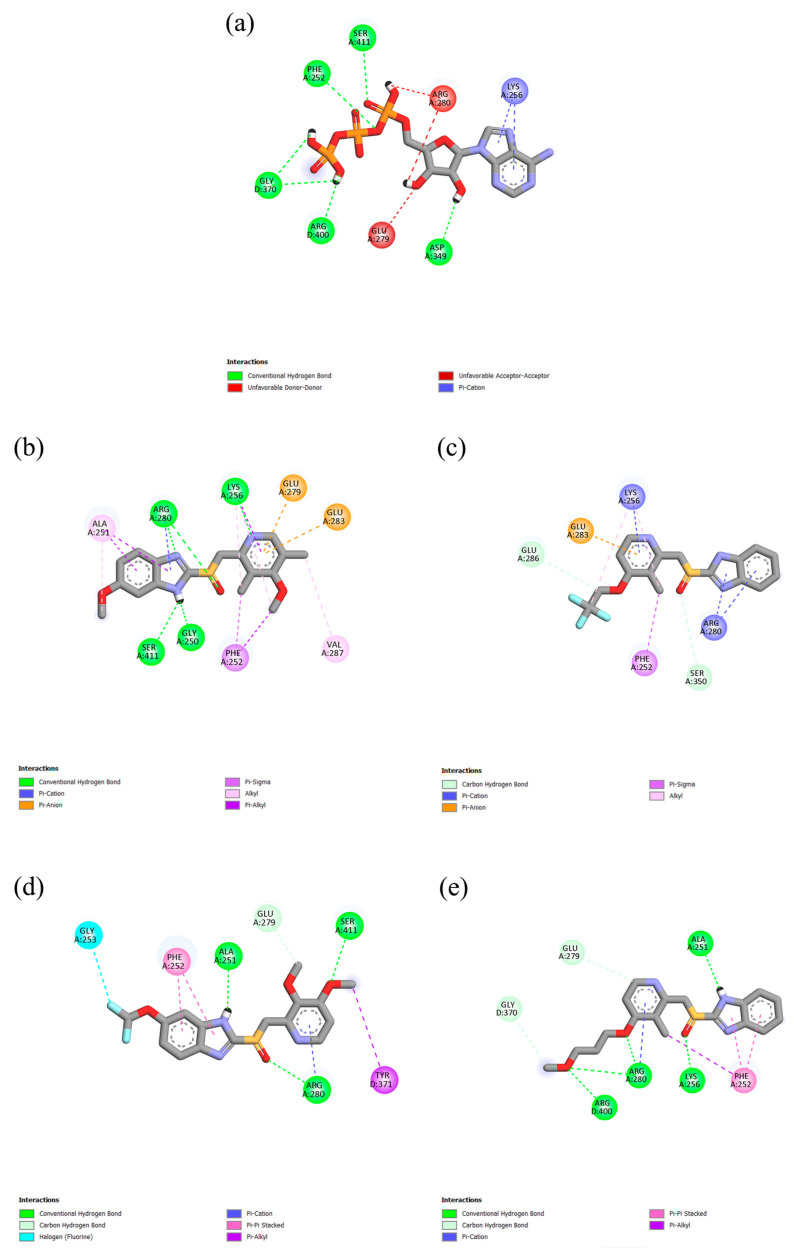
Molecular interactions of (**a**) ATP, (**b**) ESO, (**c**) DEX, (**d**) PAN, and (**e**) RAB with human V-ATPase.

**Figure 4 ijms-26-08170-f004:**
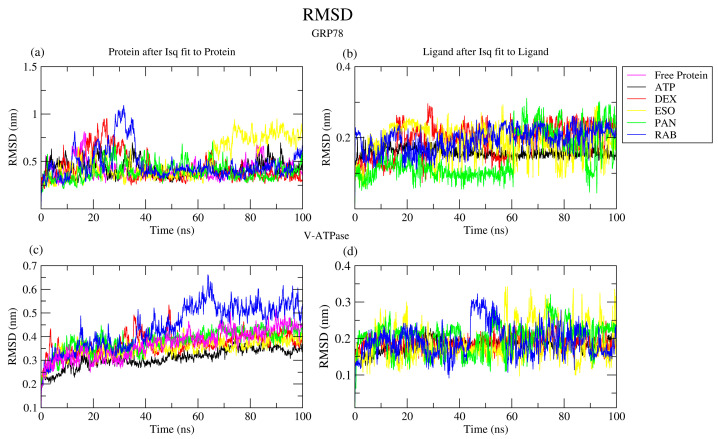
The RMSDs of the proteins (**a**) GRP78 and its complexes with the drugs DEX, ESO, PAN, RAB, and ATP and (**c**) V-ATPase and its complexes with the ligands and the RMSDs of each ligand (**b**,**d**) during the 100 ns simulations.

**Figure 5 ijms-26-08170-f005:**
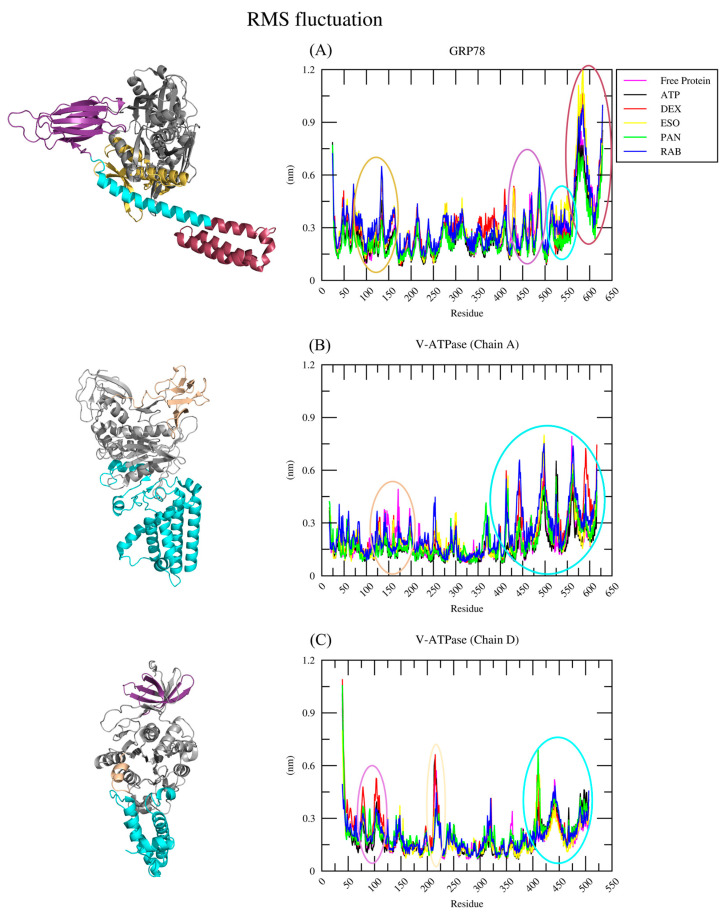
Residue RMSFs of (**A**) free GRP78 and its complexes with DEX, ESO, PAN, RAB, and ATP and of (**B**,**C**) free V-ATPase and its complexes with the ligands in chains A and D, respectively, with key regions encircled with the same color and located on their respective protein structures.

**Figure 6 ijms-26-08170-f006:**
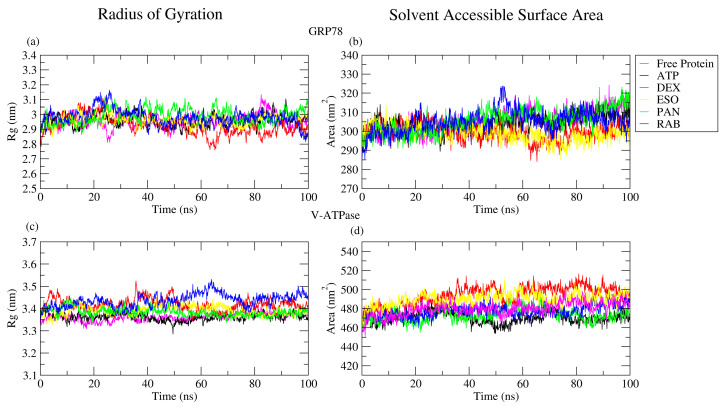
The radius of gyration and the solvent-accessible surface area of (**a**,**b**) GRP78 and its complexes with the ligands DEX, ESO, PAN, RAB, and ATP and (**c**,**d**) V-ATPase and its complexes with the ligands.

**Figure 7 ijms-26-08170-f007:**
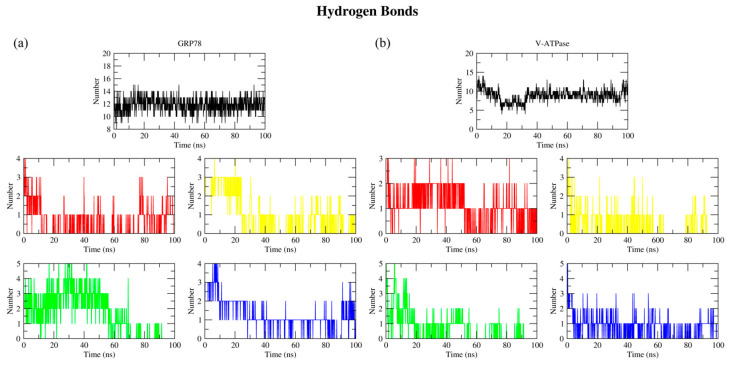
Numbers of observed hydrogen bonds in (**a**) the protein–ligand complexes of GRP78 and (**b**) V-ATPase, where black indicates ATP, red indicates DEX, yellow indicates ESO, green indicates PAN, and blue indicates RAB.

**Figure 8 ijms-26-08170-f008:**
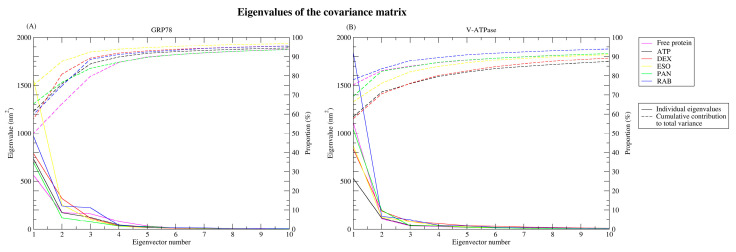
Eigenvalues and their respective accumulative percentage across eigenvectors of the (**A**) GRP78 and (**B**) V-ATPase structures.

**Figure 9 ijms-26-08170-f009:**
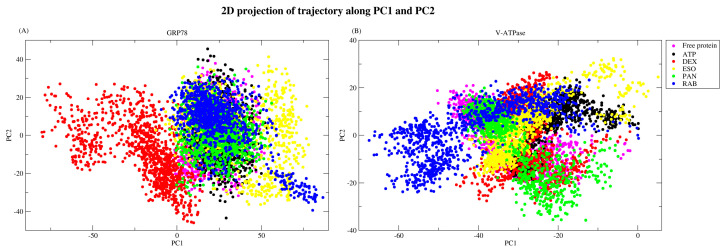
Superimposed trajectory projection of the (**A**) GRP78 and (**B**) V-ATPase structures on the PC1 and PC1 vectors.

**Figure 10 ijms-26-08170-f010:**
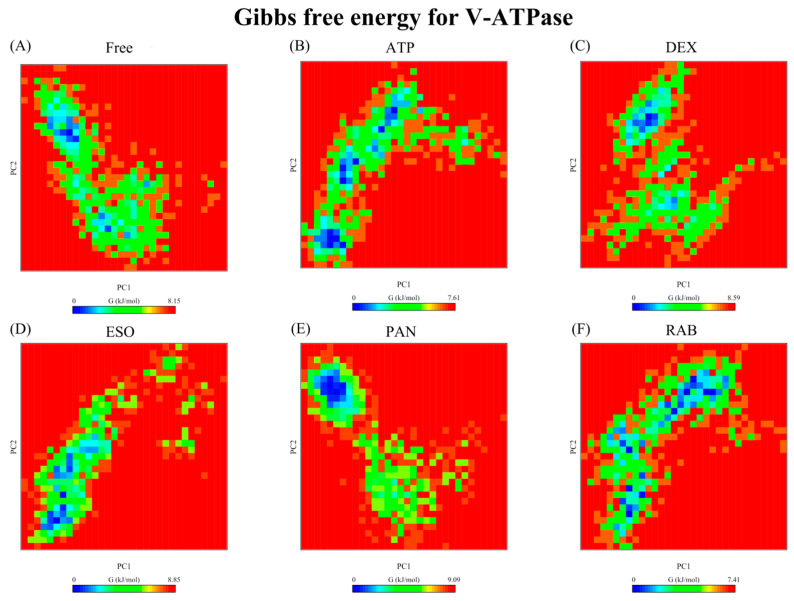
Free-energy landscapes of GRP78 structures, the (**A**) Apo protein, (**B**) ATP, (**C**) DEX, (**D**) ESO, (**E**) PAN, and (**F**) RAB complexes between PC1 and PC2.

**Figure 11 ijms-26-08170-f011:**
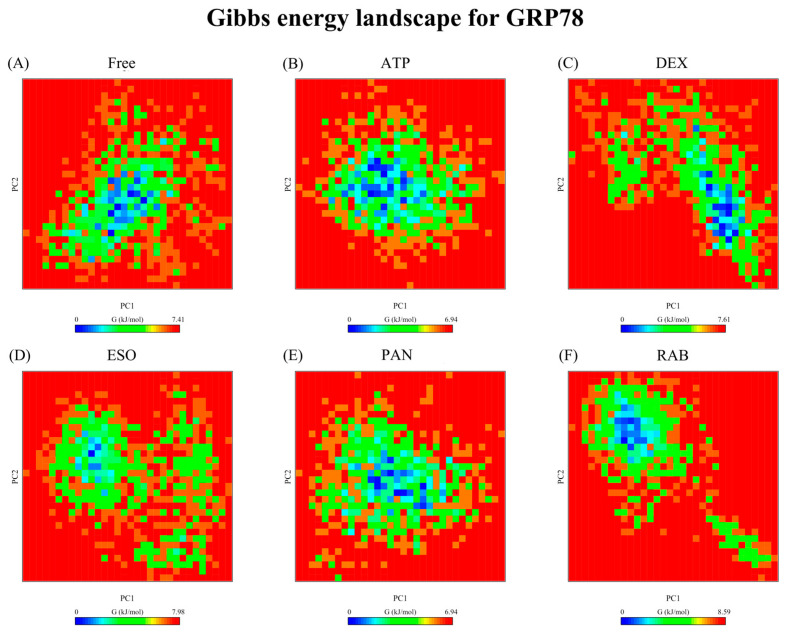
Free-energy landscapes of V-ATPase structures, the (**A**) Apo protein, (**B**) ATP, (**C**) DEX, (**D**) ESO, (**E**) PAN, and (**F**) RAB complexes between PC1 and PC2.

**Figure 12 ijms-26-08170-f012:**
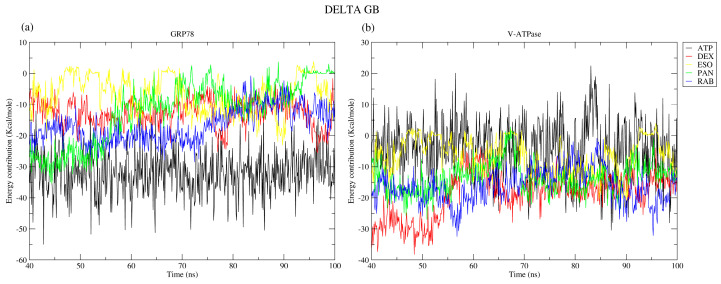
MM-GBSA analysis for (**a**) GRP78 and (**b**) V-ATPase complexes with the drugs DEX, ESO, PAN, RAB, and ATP.

**Figure 13 ijms-26-08170-f013:**
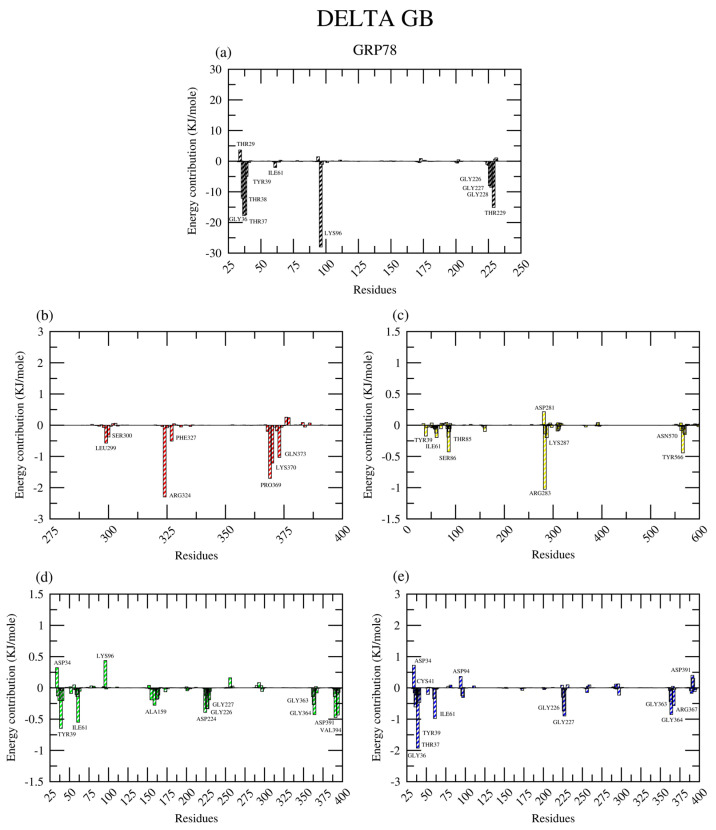
The amino acid residue contributing to the (**a**) ATP (black), (**b**) DEX (red), (**c**) ESO (yellow), (**d**) PAN (green), and (**e**) RAB (blue) complexes with GRP78, according to the MM-GBSA analysis.

**Figure 14 ijms-26-08170-f014:**
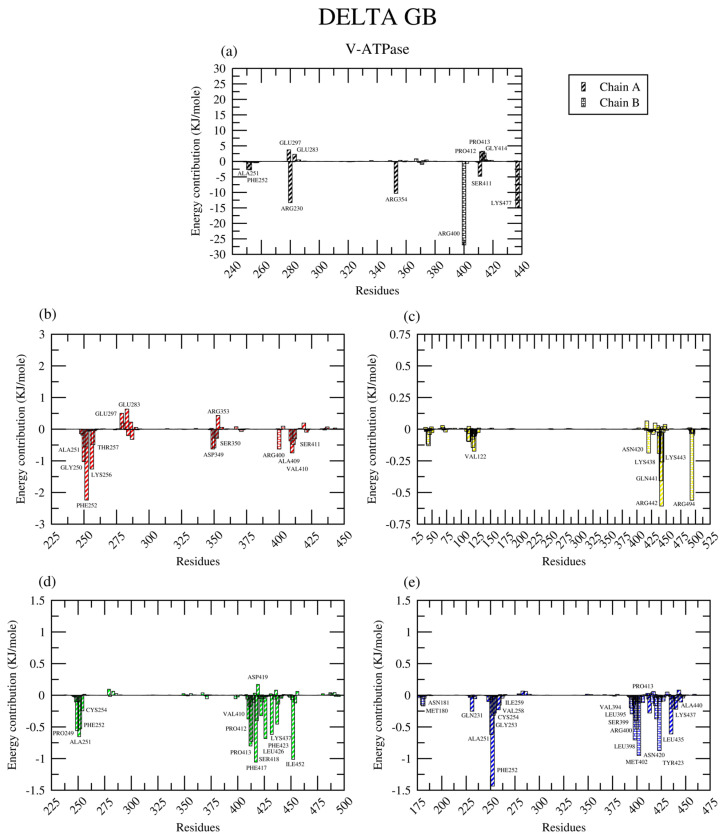
The amino acid residues contributing to the (**a**) ATP (black), (**b**) DEX (red), (**c**) ESO (yellow), (**d**) PAN (green), and (**e**) RAB (blue) complexes with V-ATPase, according to the MM-GBSA analysis.

**Table 1 ijms-26-08170-t001:** Average values of trajectory analysis of the human GRP78 and V-ATPase complexes.

Protein	Ligand	RMSD (nm)	Ligand RMSD (nm)	RMSF (nm)	RoG (nm)	SASA (nm^2^)	H-Bond
GRP78	-	~0.41	-	~0.17	~2.96	~304.84	-
	ATP	~0.42	~0.16	~0.15	~2.96	~304.43	~11.9
	DEX	~0.43	~0.20	~0.22	~2.93	~299.27	~0.62
	ESO	~0.51	~0.19	~0.20	~2.96	~298.94	~0.97
	PAN	~0.40	~0.15	~0.16	~2.99	~305.74	~1.66
	RAB	~0.47	~0.19	~0.21	~2.98	~305.21	~1.29
V-ATPase	-	~0.37	-	~0.14	~3.36	~479.68	-
	ATP	~0.31	~0.18	~0.12	~3.36	~470.65	~8.83
	DEX	~0.37	~0.19	~0.17	~3.42	~493.87	~1.08
	ESO	~0.35	~0.18	~0.18	~3.39	~473.29	~0.42
	PAN	~0.38	~0.20	~0.16	~3.38	~477.53	~0.66
	RAB	~0.44	~0.19	~0.22	~3.44	~489.77	~0.78

**Table 2 ijms-26-08170-t002:** The energy contribution of different forces on ligand binding to human GRP78 and V-ATPase, calculated using gmx MM-GBSA.

Protein	Ligand	VDWAALS	EEL	EGB	ESURF	GGAS	GSOLV	Total
GRP78	ATP	−32.64 ± 0.17	−573.43 ±1.84	579.06 ±1.78	−5.02 ±0.01	−606.07 ±1.84	574.04 ±1.78	−32.03 ±0.28
	DEX	−25.26 ± 0.18	−15.33 ±0.50	31.37 ±0.39	−3.49 ±0.03	−40.59 ±0.51	27.89 ±0.38	−12.70 ±0.18
	ESO	−13.83 ± 0.33	−5.97 ±0.32	15.08 ±0.41	−1.95 ±0.05	−19.80 ±0.55	13.13 ±0.37	−6.67 ±0.23
	PAN	−23.35 ± 0.54	−16.24 ±0.56	30.94 ±0.73	−3.36 ±0.07	−39.60 ±1.03	27.58 ±0.66	−12.02 ±0.40
	RAB	−34.61 ± 0.24	−19.35 ±0.37	42.38 ±0.37	−5.00 ±0.03	−53.96 ±0.52	37.37 ±0.35	−16.59 ±0.22
V-ATPase	ATP	−31.89 ± 0.21	−500.3 ±1.81	534.4 ±1.69	−6.35 ±0.01	−532.26 ±1.78	528.11 ±1.69	−4.16 ±0.37
	DEX	−32.58 ± 0.17	−28.44 ±0.50	46.93 ±0.37	−4.88 ±0.02	−61.02 ±0.60	42.05 ±0.36	−18.97 ±0.28
	ESO	−12.42 ± 0.40	−5.30 ±0.32	14.02 ±0.48	−1.65 ±0.05	−17.72 ±0.63	12.36 ±0.43	−5.35 ±0.23
	PAN	−25.29 ± 0.22	−16.60 ±0.46	32.20 ±0.47	−3.74 ±0.03	−41.89 ±0.57	28.46 ±0.45	−13.43 ±0.20
	RAB	−28.78 ± 0.22	−11.08 ±0.30	27.83 ±0.30	−4.16 ±0.03	−39.86 ±0.43	23.67 ±0.28	−16.19 ±0.22

All data were expressed as the mean ± standard error (SEM). All units are reported in Kcal/mol.

## Data Availability

Data is available upon request from the corresponding author.

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
