# Peer review of "Exploring the Anticancer Potential of Proton Pump Inhibitors by Targeting GRP78 and V-ATPase: Molecular Docking, Molecular Dynamics, PCA, and MM-GBSA Calculations"

_ijms, 2025, doi:10.3390/ijms26178170_

Round 1
Reviewer 1 Report
Comments and Suggestions for Authors
First of all, I congratulate the authors for such amazing research.
Although the research literature and the output have been perfectly described, I have some suggestions below that will increase the readability of the research for the readers. I also have identified some minor issues and errors present inside the manuscript.
1- Abstract needs modification and improvement.
2- Figure 2 exhibits identical labels for GRP78 and V-ATPase, suggesting a potential copy-paste error.
3- In the measurement of binding energies, a discrepancy exists between kcal/mol and KJ/mol, necessitating a preference for uniformity.
4- There are language issues.
5- Figures are not properly cited inside the text.
6- Statements like “PPIs demonstrated superior binding affinity scores for GRP78 compared to V-ATPase” would benefit from a statistical qualification of the observed difference.
7- The PCA section (2.2.5) demonstrates eigenvalue trends through graphical representation but fails to provide a comprehensive analysis.
8- Improve the conclusion.
Author Response
Dear Editor,
We sincerely thank you and the reviewers for your time, effort, and constructive feedback on our manuscript. We have carefully addressed all comments and incorporated the necessary revisions to enhance the quality and clarity of the work. Below, we provide a detailed, point-by-point response to each of the reviewers' comments. All changes made in the revised manuscript have been highlighted.
We hope that the revised version meets the expectations, and we remain available to address any further questions or suggestions.
Reviewer #1
First of all, I congratulate the authors for such amazing research.
Although the research literature and the output have been perfectly described, I have some suggestions below that will increase the readability of the research for the readers. I also have identified some minor issues and errors present inside the manuscript.
Author reply: Thank you for your valuable feedback and for acknowledging the effort put into this study.
1- Abstract needs modification and improvement.
Author reply: Thank you for your valuable feedback. We amended the abstract in the revised version.
2- Figure 2 exhibits identical labels for GRP78 and V-ATPase, suggesting a potential copy-paste error.
Author reply: Thank you for your valuable feedback. We amended the labeling of Figure 2 and Figure 3 in the revised version.
3- In the measurement of binding energies, a discrepancy exists between kcal/mol and KJ/mol, necessitating a preference for uniformity.
Author reply: Thank you for your valuable feedback. We corrected this issue in the revised version.
4- There are language issues.
Author reply: Thank you for your valuable feedback. We revised the manuscript usin quillbot and corrected any errors in the revised version.
5- Figures are not properly cited inside the text.
Author reply: Amended in the revised version.
6- Statements like “PPIs demonstrated superior binding affinity scores for GRP78 compared to V-ATPase” would benefit from a statistical qualification of the observed difference.
Author reply: Amended in the revised version.
7- The PCA section (2.2.5) demonstrates eigenvalue trends through graphical representation but fails to provide a comprehensive analysis.
Author reply: Thank you for your valuable feedback. Amended in the revised version.
8- Improve the conclusion.
Author reply: Amended in the revised version.
Reviewer 2 Report
Comments and Suggestions for Authors
The article presents an interesting approach regarding the potential repurposing of classical drugs for cancer therapy. The topic is relevant, and the findings are promising; however, there are several points that could be addressed to make the work more comprehensive:
-
Inclusion of Reference Ligands for Docking
In the docking study, it would be advisable to include one or more reference ligands. Although the protein structures used do not have co-crystallized ligands, experimentally validated ligands can be identified in the literature and included in the ligand pool. This would allow a meaningful comparison and contextualization of the docking results, as binding energies alone represent relative values rather than absolute measures. -
Improved Visualization of Interactions in Figure 2
In Figure 2, certain interaction types are displayed using the same color. This appears to be the default output of Discovery Studio. For clarity and better interpretation, it is recommended to adjust the settings so that each type of molecular interaction is represented by its own distinct color. -
Consideration of the Membrane Environment for V-ATPase Simulations
V-ATPase is a transmembrane protein that operates in the lipophilic environment of the lipid bilayer in the cell membrane. Performing molecular dynamics simulations in an aqueous environment may not fully reflect the physiological conditions of this enzyme. It would be appropriate to consider, or at least discuss, the possibility of running simulations in a lipid bilayer environment to better represent its native state and potentially yield more accurate insights into ligand binding. -
Strength of the Final Conclusion
Considering that the reported RMSD values are relatively high, the final statement — “According to our findings, RAB and DEX are potential candidates as inhibitors against GRP78 and V-ATPase, respectively” — may be somewhat bold. While RAB and DEX could indeed be considered promising molecules for experimental testing as potential inhibitors, presenting them already as well-defined inhibitor candidates might be premature without further in vitro or in vivo validation.
Author Response
Reviewer #2
The article presents an interesting approach regarding the potential repurposing of classical drugs for cancer therapy. The topic is relevant, and the findings are promising; however, there are several points that could be addressed to make the work more comprehensive:
- Inclusion of Reference Ligands for Docking
In the docking study, it would be advisable to include one or more reference ligands. Although the protein structures used do not have co-crystallized ligands, experimentally validated ligands can be identified in the literature and included in the ligand pool. This would allow a meaningful comparison and contextualization of the docking results, as binding energies alone represent relative values rather than absolute measures.
Author reply: Thank you for your valuable feedback. We already used a reference compound (ATP) in our study for both GRP78 and V-ATPase.
- Improved Visualization of Interactions in Figure 2
In Figure 2, certain interaction types are displayed using the same color. This appears to be the default output of Discovery Studio. For clarity and better interpretation, it is recommended to adjust the settings so that each type of molecular interaction is represented by its own distinct color.
Author reply: Thank you for your valuable feedback. We amended Figure 2 and Figure 3 in the revised version.
- Consideration of the Membrane Environment for V-ATPase Simulations
V-ATPase is a transmembrane protein that operates in the lipophilic environment of the lipid bilayer in the cell membrane. Performing molecular dynamics simulations in an aqueous environment may not fully reflect the physiological conditions of this enzyme. It would be appropriate to consider, or at least discuss, the possibility of running simulations in a lipid bilayer environment to better represent its native state and potentially yield more accurate insights into ligand binding.
Author reply: Thank you for your valuable feedback. We are working on part of the V-ATPase which localize in the cytoplasm (chains A and D of the V1 complex), therefore we didin’t used membranes in our system.
- Strength of the Final Conclusion
Considering that the reported RMSD values are relatively high, the final statement — “According to our findings, RAB and DEX are potential candidates as inhibitors against GRP78 and V-ATPase, respectively”— may be somewhat bold. While RAB and DEX could indeed be considered promising molecules for experimental testing as potential inhibitors, presenting them already as well-defined inhibitor candidates might be premature without further in vitro or in vivo validation.
Author reply: Thank you for your valuable feedback. We amended the conclusion section in the revised version and highlighted the changes made.